# Trypanosome Infections and Anemia in Cattle Returning from Transhumance in Tsetse-Infested Areas of Cameroon

**DOI:** 10.3390/microorganisms11030712

**Published:** 2023-03-09

**Authors:** Oumarou Farikou, Gustave Simo, Flobert Njiokou, Ginette Irma Kamé Ngassé, Martin Achiri Fru, Anne Geiger

**Affiliations:** 1Faculty of Health Sciences, Department of Biomedical Sciences (BMS), University of Bamenda, Bambili P.O. Box 39, Cameroon; 2Molecular Parasitology and Entomology Unit, Department of Biochemistry, Faculty of Sciences, University of Dschang, Dschang P.O. Box. 67, Cameroon; 3Faculty of Sciences, Department of Animal Biology and Physiology, University of Yaounde I, Yaoundé P.O. Box 812, Cameroon; 4Institute of Medical Research and Plant Medicinal Studies (IMPM), Ministry of Scientific Research and Innovation, Yaoundé P.O. Box 13033, Cameroon; 5Special Mission for Tsetse Fly Eradication, Ministry of Livestock, Fisheries and Animal Industries, Ngaoundere P.O. BOX 812, Cameroon; 6Institut de Recherche Pour le Développement, UMR INTERTRYP, 34398 Montpellier, France

**Keywords:** cattle trypanosomiasis, transhumance, packed-cell volume, trypanosome species, immunological test, PCR-based method, Cameroon

## Abstract

The objective of this work was to assess the anemic status and the use of an immunological test and PCR-based methods to determine the infection rates of trypanosomes species. Transhumance aims to provide cattle with greener pastures and greater water resources than in the Djerem region during the dry season. Two criteria were used to assess the health status of the animals, the prevalence of trypanosomiasis and the level of anemia. In addition, we have evaluated the effectiveness, in trypanosomiasis detection, of the Very Diag Kit (CEVA Santé animale), a Rapid diagnosis test (RDT) based on immunological identification of *T. congolense* s.l. and *T. vivax*, responsible for AAT. Four trypanosome species (*Trypanosoma congolense* savannah type (Tcs), *T. congolense* forest type (Tcf), *T. brucei* s.l. (Tbr) and *T. vivax* (Tvx)) were identified in cattle sampled in four villages. The overall infection rate determined by PCR (68.6%) was much higher than those generally reported in cattle from the Adamawa region (35 to 50%). Infections (including mixed infections) by Tc s.l. (Tcs + Tcf) were predominant (45.7%). The infection rates were also determined using the Very Diag Kit allowing us to identify Tc s.l. and Tvx in the field in less than 20 min. This method provided, for the global infection, a higher rate (76.5%) than that determined by PCR (68.6%), although it is supposed to be less sensitive than PCR. Tc s.l. infection rate (37.8%) was similar to that (38.8%) determined by PCR (Tcs + Tcf single infections). In contrast, the prevalence of Tvx single infections measured by RDT (18%) was nearly two-fold higher than that (9.4%) measured by PCR. Thus, further comparative analyses seem to be needed in order to more accurately assess the sensitivity and specificity of the Very Diag test under our conditions of use on blood samples. The mean PCVs in trypanosome-infected as well as in uninfected cattle were below 25%, the threshold below which an animal is considered anemic. Our study shows that cattle return from transhumance in poor health. It raises questions about its real benefit, especially since the herds are themselves likely to become vectors of trypanosomiasis and possibly of other diseases. At least, effective measures have to be undertaken to treat all cattle coming back from transhumance.

## 1. Introduction

In Cameroon, ruminants are usually reared according to three traditional extensive grazing systems:Nomadism: Herders do not have a fixed home and move throughout the year in search of water and fodder for their animals;Transhumance: Herders have a permanent residence but move seasonally, depending on the rainy and dry seasons, in search of better grasslands and water availability;Sedentary breeding: The breeders keep their cattle close to the village where they live. However, during the dry season, they sometimes entrust part of their herd to a transhumant herder. The sedentary population is typically made up of farmers–breeders, with livestock being a complement to crop production.

The climate changes that have taken place over the past two decades have led to difficulties in accessing pastures. For herders from the Adamaoua region, one of the alternatives to overcome this situation is to practice transhumance [1,2]. However, in sub-Saharan Africa, livestock transhumance often plays an important role in the dynamics of multiple infectious livestock diseases [3,4].

Fulani herders represent the largest community of the local population and own more than 90% of the cattle herds.

In most sub-Saharan countries where animal African trypanosomiasis is endemic, most breeders practice transhumance, in which they move seasonally and every year to other regions in the search of pastures with better availability of grasslands and water. Essential when the climatic conditions become unfavorable and the food availability scarce for cattle breeding [5,6], searching new grazing areas through the transhumance phenomenon is of great importance to the nutrition and the health of cattle of each herd. During this transhumance phenomenon, herds cross various bioecological settings including national parks and tsetse-infested areas. Through their passage in such areas, herds are exposed not only to trypanosome infections and AAT but also to other vector-borne diseases. To fight against AAT, insecticides are generally spread on cattle of each herd before the transhumance in order to kill tsetse flies and prevent the transmission of trypanosome species [4,6]. However, the spread of insecticides is not performed when the herds are returning from the transhumance. As cattle spend several weeks in some tsetse-infected areas on their way back after the transhumance, they are exposed to biological (tsetse flies) and mechanical (*Stomoxinae* and *Tabanidae*) vectors of trypanosomes and therefore, can acquire trypanosome infections. Such infections may have some influences on the anemia status of cattle. Up till now, the trypanosome infections as well as the anemic status of cattle returning from transhumance in tsetse-infested areas have been overlooked by many scientists. The sampling of cattle a few hours after their arrival in the quarantine zone is a novelty in our context.

This study was designed to assess the anemic status and use immunological tests and PCR-based methods to determine the infection rates of trypanosome species in cattle returning from transhumance in tsetse-infested regions of Cameroon.

## 2. Methods

### 2.1. Sites of the Study

The Djerem Division is located to the south of Adamawa region, and it covers about 13,283 km^2^. It is situated between 6.10° and 7.0° north and 12.00° and 13.5° east. The climate is of the Sudanian type with two seasons: One dry season (from October to March) and one rainy season (from April to September). Its hydrographic network is dense, and the main rivers are Djerem and Meng rivers. The vegetation is characterized by savannah. One part of the Djerem division includes the Mbam and Djerem reserve (Figure 1), a protected area where several species of wild animals are living.

### 2.2. Blood Sample Collection in Cattle

Blood samples were collected in May 2020 from cattle in four villages (Bamti Mbam, Djoundé, Ngaoundal, and Ngatt) located in the Djerem Division, Adamawa Region, Cameroon. The cattle were selected from herds returning from 5 months of transhumance in the locality of Yoko located in the Center region at about 202 km from the four villages. During the transhumance, cattle cross different tsetse-infested regions including the Mbam and Djerem animal reserves.

Before blood collection, the objective of the study was explained to the local authorities, local veterinarians, and inhabitants of different villages. After approval, shepherds were asked to lead their cattle at the animal park. In each village or sampling site, at least 100 cattle 3 to 12 years old were randomly selected from several herds. From each selected animal, about 5 mL of blood was collected into EDTA-coated tubes. Bleeding was performed in the jugular vein after specific contention of the cattle with a solid rope. In the field, an aliquot of each blood sample was used to search for trypanosome infections using immunological test (Very Diag Kit) provided by CEVA Santé animale (France). Another aliquot was used to estimate the anemic status of cattle. The remaining blood sample was stored at 4 °C for molecular detection of trypanosome infections.

A total of 405 blood samples were collected in cattle from four villages (Table 1): 100 cattle from Bamti Mbam, 105 cattle from Djoundé, 100 cattle from Ngaoundal, and 100 cattle from Ngatt. All these cattle belonged to Zebu Gudali breed. These cattle included 100 males and 305 females.

### 2.3. Assessment of the Anemic Status of Cattle

The packed cell volume (PCV) approach was used to assess the cattle’s anemic status. This was achieved by introducing 70 µL of each blood sample into heparinized capillary tube which extremities were sealed with soap. After centrifugation at 11,000 rpm for 5 min, the PCV was determined using a hematocrit reader. An animal was considered anemic when its PCV was below 25% [7].

### 2.4. Detection of Trypanosome Infections Using Immunological Test of the Very Diag Kit

Trypanosome infections caused by either *Trypanosoma congolense* or *T*. *vivax* were investigated using the Very Diag Kit. Briefly, one drop of blood was introduced into the cassette, followed immediately by a drop of kit solution was immediately introduced into the cassette. The cassette was then incubated at room temperature for 5 min. The mixture (blood and kit’s solution) begins to migrate upwards, and the control band appears first. Then, the migration continues until the sample is either positive (development of a *T. congolense* specific or/and *T. vivax* specific band), or negative (absence of a second band).

### 2.5. Detection of Trypanosome Infections Using PCR-Based Method

#### 2.5.1. DNA Extraction from Cattle Blood

DNA was extracted from blood samples using DNeasy Tissue Kit (QIAGEN-France). One mL of blood sample was mixed with 1 mL of sterile water and vortexed before centrifuging at 14,000 rpm for 10 min. The supernatant was discarded and the pellet containing the parasites was re-suspended in 200 µL of PBS. From the re-suspended pellets, DNA was extracted following the manufacturer’s instructions. DNA extracts were stored at −20 °C until use.

#### 2.5.2. Molecular Identification of Different Trypanosome Species in Cattle Blood

Different trypanosome species were identified by amplifying the internal transcribed spacer 1 (ITS1) fragment of ribosomal DNA. The amplification reaction was performed in two PCR rounds as described by Farikou et al. (2022) [8].

At this step, *T. congolense* (~700 bp), *T. brucei* s.l. (~480 bp) and *T. vivax* (Tvx) (~250 bp) can easily be identified. For the identification of different types of *T. congolense*, specific primers were subsequently used.

#### 2.5.3. Molecular Identification of *T. congolense* Savannah and Forest “Types”

The identification of the different types of *T. congolense* was performed on all blood samples that showed, after amplification of trypanosome ITS1, DNA fragments of ~700 bp corresponding to the expected size of *T. congolense* species. To discriminate between the two types of *T. congolense*, specific primers for *T. congolense* forest type [9] and *T. congolense* savannah type [10] were used to amplify DNA fragments for each of these trypanosomes. PCR reactions were performed as described by Farikou et al. (2022) [8]. The amplified products were resolved by electrophoresis on 2% agarose gel, which was subsequently stained by ethidium bromide and visualized under UV light.

### 2.6. Data Analysis

Pearson’s chi-square test (χ^2^) was used to compare cattle infection rates between villages. All the statistical tests were performed with the PASW Statistics 25 software (SPSS Inc., Chicago, IL, USA), and the threshold of significance was set at *p* value below 0.05. The mean PCV values of cattle from different villages were compared using ANOVA test. Association between the mean PCV values in cattle and trypanosome infections was investigated using the Student test. The kappa values with 95% CIs were used to determine the concordance between VDT and PCR. These values were interpreted according to the classification of Landis and Koch (1977) [11].

## 3. Results

### 3.1. Trypanosome Infections Revealed by Immunological Test: Very Diag Test

Table 1 shows the results recorded on trypanosome infection rates (single and mixed infections) using the Very Diag test.

Out of 405 sampled cattle, 310 were positive, i.e., a global infection rate of 76.5%; 37.8% for *T. congolense*, 18.0% for *T. vivax,* and 20.7% for mixed infection. The global infection rate varied slightly between cattle from the different villages: 79% in cattle from Bamti Mbam, 73.3% in those from Djoundé, 75% in those from Ngaoundal, and 79% in cattle from Ngatt. Comparing these infection rates, no significant difference (χ^2^ = 1.408; *p* = 0.704) was recorded between villages. Regarding single infection, the rate in Ngatt, Ngaoundal, and Djoundé was nearly similar (52, 53 and 52.4%, respectively), whereas that from Bamti Mbam (66%) were quite different; nevertheless, the differences over the four villages were not significant. In contrast, differences in Tc s.l. infection over the four villages (49% in Bamti Mbam vs. 33% in Djoundé and Ngatt, and 36% in Ngaoundal) were nearly significant (*p* = 0.06). Regarding the mixed infections, their infection rates varied between villages: 13%, 20.9%, 22%, and 27%, respectively, in cattle from Bamti Mbam, Djoundé, Ngaoundal, and Ngatt; however, the differences were not significant (χ^2^ = 6.128; *p* = 0.106).

### 3.2. Trypanosome Infections Revealed by Molecular Methods

Four different trypanosomes including *T. congolense* savannah type, *T. congolense* forest type, *T. vivax,* and *T. brucei* s.l. were identified by PCR (Table 2). One may note that “*T. brucei* s.l.” (referred to as Tbr in the tables and figures) could be a mixture of trypanosomes belonging to the subgenus *Trypanozoon*; however, in the context of the present study they are most probably *T. brucei brucei*—see the discussion section.

Out of the 405 Zebu Gudali cattle, 278 were infected with at least one trypanosome species: Giving an overall infection rate of 68.64%. A total of 220 (54.32%) of them were singly infected while 58 (14.32%) had mixed infection. The overall trypanosome infection rate was higher in Djoundé (75.23%) followed by Ngaoundal (72%), Bamti Mbam (69%), and Ngatt (58%). Comparing the overall trypanosome infection rates, significant differences (χ^2^ = 7.91; *p* = 0.05) were recorded between villages. While the differences, between the four villages, in the prevalence of single infections are not significant (*p* = 0.85), those of the prevalence of mixed infections were highly significant (*p* = 0.004).

The infection rates of *T. congolense* savannah type (Tcs) were higher across the four villages (mean infection rate: 31.8%; prevalence differences between the sites are not significant (*p* = 0.31). Infection rates caused by *T. vivax*, *T. congolense* forest type, and trypanosomes of the subgenus *Trypanozoon*, were 9.4%, 6.9%, and 6.2%. Comparing these infection rates, no significant difference was recorded between villages for *T. congolense* forest type (χ^2^ = 4.14; *p* = 0.24), *T. congolense* savannah type (χ^2^ = 3.58; *p* = 0.31), and *T. vivax* (χ^2^ = 6.32; *p* = 0.09). However, for trypanosomes of the subgenus *Trypanozoon*, these infection rates differ significantly (χ^2^ = 8.67; *p* = 0.03) between the four villages.

Regarding mixed infections, six combinations including each of two different trypanosome species were identified, and one included three species. The six double infections involved, respectively, *T. congolense* savannah and *T. brucei* s.l. trypanosome of the subgenus *Trypanozoon* (4.2% global infection rate), *T. congolense* savannah type and *T. vivax* (2.7%), *T. vivax,* and *T. brucei* s.l. (2.5%), *T. congolense* savannah type and *T. congolense* forest type (1.5%); *T. congolense* forest type and *T. brucei* s.l. (0.98%), *T. congolense* forest type, and *T. vivax* (0.98%). The triple infections included *T. congolense* savannah type, *T. congolense* forest type, and *T. brucei* s.l. (1.5%). Among these mixed infections, only the differences in infection rates of those due to Tcs/Tbr and of Tvx/Tbr were significant (*p* = 0.03 and 0.02, respectively). It should be noted that the infection rates were highly variable both: (i) when one considers, for a given site, the prevalence of the seven combinations of species involved in the infections, and (ii) when one considers, for a given type of mixed infection, its prevalence in the different villages. For example, all the seven types of mixed infection were recorded in cattle from Djoundé (prevalence varying from 0.95 to 7.6%), while only two were identified in Ngatt (Tcs/Tbr: 3%; Tcs/Tvx: 2%). Mixed infections involving *T. congolense* savannah and *T. vivax* were identified in cattle from all villages (infection rates varying from 2 to 4%), while those involving Tcs and Tbr were detected only in cattle from Bamti Mbam (1%) and Djoundé (2.8%).

### 3.3. Comparison between the Infection Rates Recorded by Immunological Tests (Very Diag Kit) and PCR-Based Method

For this comparison, the results of the PCR-based method detecting Tb*r* trypanosomes of the subgenus *Trypanozoon* were not taken into account since the Very Diag test was not designed to detect it. This comparison targeted cattle that were positive or negative for either *T. vivax* or *T. congolense* forest and/or savannah type.

The comparison was made on the results recorded for simple infections (Table 1 and Table 2). Concerning the detection of *T. congolense* (sensu lato), the infection rate recorded by VDT, 37.8%, was quite similar to that obtained by PCR, 38.8% (31.85% Tcs + 6.91% Tcf). In contrast, regarding *T. vivax*, the infection rate recorded by the VDT (18%) was nearly two-fold higher than that obtained by the PCR method (9.4%). The overall infection rate of *T. congolense* and *T. vivax* recorded by VDT and PCR, (55.8% and 48.6%, respectively), reflects this discrepancy (Figure 2).

### 3.4. PCV Values or Anemia Status of Cattle from the Four Villages

Table 3 summarizes the results of cattle anemia in the four villages. The mean PCV value for all cattle was 23.15 which means the cattle were not in good health since animals whose PCV is less than 25 are considered to be anemic. Cattle from Ngaoundal had the highest PCV values of 24.1 and those from Bamti Mbam had the lowest PCV value of 22.4. For cattle from Djounde and Ngatt, their PCV values were, respectively, 23.6 and 22.5. Comparing the values of PCV, significant differences (χ^2^ = 3.8; *p* = 0.01) were recorded between cattle from the four villages. Cattle from the four villages were anemic.

### 3.5. Cattle Anemia: Effect of Trypanosome Infection on Cattle Anemia across the Sampling Sites

For all trypanosome infections, Figure 3 presents the results recorded when the infection was identified using the Very Diag Kit (Figure 3A) and the PCR-based method (Figure 3B). Both infected and uninfected cattle were anemic; only uninfected cattle from Ngaoundal had a mean PCV slightly higher than 25. PCV values were slightly lower in infected cattle diagnosed by immunological tests (Very Diag Kit) compared to the PCR-based method.

The second step consisted of investigating the possible effect of different types of infections (here only a single infection identified using a molecular approach was considered) on the level of anemia in cattle. Results are shown in Appendix A: Tcs-infected cattle; Appendix A: Tcf-infected cattle; Appendix A: Tvx-infected cattle; Appendix A: Tbr s.l.-infected cattle).

From a general point of view, infection by a given trypanosome species or another one had no crucial effect on the severity of anemia in cattle from the different villages. This is particularly true when cattle were infected with Tcs; their PCV was shown to be very similar to those recorded when the overall cattle infections are considered. This concordance may be related to the fact that Tcs infections represented nearly 60% of overall infections identified in cattle. Similarly, regarding anemia of Tcf-infected cattle, except for those from Ngatt where PCV was greater than 25.

Levels of anemia in cattle infected with either *T. vivax* or trypanosomes of the subgenus *Trypanozoon* were more scattered with PCV remaining either slightly below or above the threshold of PCV = 25. The latter case was observed in three villages (Bamti Mbam, Ngaoundal, and Ngatt) in Tvx positive cattle and, in the village of Djoundé in Tbr positive cattle. It should, however, be noted that, in this latter case, only three animals were identified (PCR) as being infected with Tbr.

## 4. Discussion

Although transhumance is a common practice carried out by farmers in most sub-Saharan countries, its impacts on trypanosome infections and anemia have been overlooked by many scientists. This study aimed to fill this gap by determining trypanosome infections and the anemic status in cattle returning from transhumance—are they better or worse compared to the average situation of the herds in the Djerem region? Thus, sampling was performed a few hours after their arrival in the quarantine zone in order to avoid the effect of the local environment on trypanosomiasis rates and/or on the diversity of infecting species, and on the level of anemia.

In the present study, we used an immunological diagnostic test, the Very Diag Kit, which was designed using TvGM6 recombinant protein and TcCB1 protein as antigens [12] to detect specifically *T. vivax* [13] and *T. congolense* [14], respectively. Compared to the PCR-based methods, it has the advantage of being easy to perform, provides results within 20 min, and can be carried out in the field or at the point of care. Immunological tests, however, have limitations, depending on the antigens they use, in terms of sensitivity and specificity in the identification of infecting trypanosomes. Thus, for example, Lumbala et al. (2018) [15] working on human African trypanosomiasis (HAT) in RDC compared with each other and with CATT (card agglutination test for trypanosomiasis) the accuracy of two RDTs: SD Bioline HAT (using natural antigens) and the more recently developed SD Bioline HAT 2.0 using recombinant antigens. SD Bioline showed the best sensitivity (71.2%) followed by CATT (62.5%) and SD Bioline HAT (59%). In terms of specificity, the three tests were roughly comparable: CATT (99.2%), SD Bioline HAT (98.9%), and SD bioline HAT 2.0 (98.1%). Similarly, Lopez-Albizu et al. (2020) [16] working on Chagas disease (caused by *T. cruzi*) in Argentina used two RDTs: Ab standard Diagnosis SD Bioline and Check Chagas Wiener Lab; they displayed, respectively, 97.2% and 93.4% sensitivity, and 91.7% and 99.1% specificity. Furthermore, many epidemiological studies on various populations using RTDs specific for Chagas disease have revealed a wide variability in their performance with sensitivity values ranging from 33% to 100% and specificity ranging from 94% to 99.9% [17,18]. Moreover, the sensitivity of the tests also seems to vary according to the nature of the samples; thus, the sensitivity of tests performed on serum may vary from 96 to 100% [17,19,20,21] while they can vary from 96 to 62.5% whole blood samples is tested [21,22]. As a consequence, Lopez-Albizu et al. (2020) [16] consider the results recorded using RDTs must be confirmed by other more specific serological tests or by PCR.

The overall of infection rate recorded in cattle by PCR upon their return from transhumance was 68.64%. This infection rate was higher than that recorded in cattle, for example, by Oumarou et al. (2022)—40.8% [23], Paguem et al. (2008)—53.2% [24], Kame Ngasse et al. (2018)—34.8% [25], Ngomtcho et al. (2017)—40% [26], who also investigated cattle trypanosomiasis in the Adamawa Region. Surprisingly, using the VDK approach, although considered less sensitive than PCR, the prevalence was 76.54%. The difference between the prevalence obtained by PCR and that recorded when using the Very Diag test is large in as much the latter test is supposed not to take into account *T. brucei* infections.

Regarding single infections, Tcs infections (31.8%) were predominant as usual in AAT, while the prevalence of Tcf infections was 6.9%; thus, the *T. congolense* s.l. infection rate was 38.7%. This result is consistent with that, 37.8%, provided by the Very Diag test. In contrast, the infection rate (18%) of *T. vivax* provided by Very Diag was two-fold higher than the one (9.4%) obtained with PCR-based methods. This discrepancy could only be explained if we consider that the immunological test is not totally specific; TvGM6 protein or a structurally very closely related protein may be present in other parasites, and in this case in the blood of some of the selected cattle. Results of this study showing a divergence in the infection rates recorded with immunological tests and PCR-based methods highlight the need to carry out additional studies aiming to assess the specificity of the Very Diag test on different biological fluids from cattle of endemic and non-endemic areas for African trypanosomiasis, but also on samples harboring pathogens that are often co-endemic with trypanosomes. In addition, in the future other tools currently under development may be available [27].

In addition to discriminating *T. congolense* into *T. congolense* forest and *T. congolense* savannah, PCR-based methods allowed identifying trypanosomes of the subgenus *Trypanozoon* (Tbr s.l.) with an overall infection rate of 15.4%. Compared to the Very Diag test, the results of this study confirm not only the capacity of PCR-based methods to discriminate trypanosome species but also the high specificity of PCR in the identification of different trypanosome species. Despite the higher specificity of this PCR-based method, the differentiation of trypanosomes of the subgenus *Trypanozoon* that include *T. b. rhodesiense*, *T. b. brucei*, *T. b. gambiense*, *T. evansi,* and *T. equiperdum* remains a challenge because no single test is able to unequivocally distinguish these trypanosome species [28]. Nevertheless, as the sampling sites were located in not endemic areas for human African trypanosomiasis, the possibility of having *T. b. gambiense* or *T. b. rhodesiense* infections in cattle must be excluded. The presence of *T. equiperdum* infections is also unlikely because this parasite is an equine trypanosome transmitted via coitus. Results of this study suggest that cattle carrying trypanosomes of the sub-genus *Trypanozoon* could be infected either by *T. evansi* or *T. b. brucei* or a combination of these two trypanosome subspecies.

The mean PCV values in infected and uninfected cattle from all the four sampling sites were less than 25%; indicating that most cattle returning from transhumance were anemic. These results indicate that cattle returning from transhumance must be treated for anemia by providing appropriate drugs and a better diet to improve the rate of hematocrit in each animal. However, the low mean PCV values recorded in cattle returning from transhumance in tsetse-infested regions cannot be related to the overall trypanosome infections or to specific trypanosome species because no significant difference was recorded in the mean PCV values between infected and uninfected cattle. Although anemia is a common feature of AAT, the fact that no correlation was recorded between trypanosome infections and the anemic status of cattle is difficult to explain. Nevertheless, it is important to point out that animals can become emaciated due to other concurrent parasitic infections such as helminth infections.

Finally, the recorded results about the trypanosomiasis rate and the overall level of anemia revealed a rather degraded health status of the cattle on their return from transhumance. Moreover, as cattle were sampled just when they came back from transhumance, the long travel distance of 50 to 200 km may have some effects on the PVC value in each animal. Indeed, depending on herders, some of them travel directly from the transhumance sites to their destination without resting periods while others observed multiple resting periods in grazing areas [4]. Compared to herds observing resting periods during their return from transhumance, the PCV values will likely be more affected in cattle moving directly from transhumance sites to the destination because these cattle did not have enough time to recover during their trips. However, multiple grazing locations increase the exposure of herds to geographically or seasonally abundant diseases [29]. Remarkably, all herds crossed the Djerem Animal Park hosting abundant and diverse wildlife species that are reservoirs for different trypanosome species [30,31,32]. In addition to that, a large diversity of biting insects including biological (tsetse fly species) and mechanical vectors of trypanosomes have been reported in this park [33]. This environment offers conditions for trypanosome transmission in cattle passing across this park. Results of this study suggest that although the transhumance offers to farmers the possibility to find greener pastures and water resources for their herds, this phenomenon exposes cattle to animal African trypanosomiasis because these animals can be bitten by biological and mechanical vectors of trypanosomes during their passage in tsetse-infested areas and as such, transhumant herds may potentially contribute to the circulation and spread of parasites and vector-borne diseases [4,34].

## 5. Conclusions

This study showed that cattle returning from transhumance harbored high infection rates of trypanosome infections. They are therefore in poor health condition. The results of this study highlighted the need to control trypanosome infections in cattle herds returning from the transhumance, especially from tsetse-infested areas. All cattle returning from transhumance in such areas could be treated with trypanocides to improve their health status and avoid the propagation of trypanosome transmission. At least, effective measures have to be taken to treat all cattle coming back from transhumance.

## Figures and Tables

**Figure 1 microorganisms-11-00712-f001:**
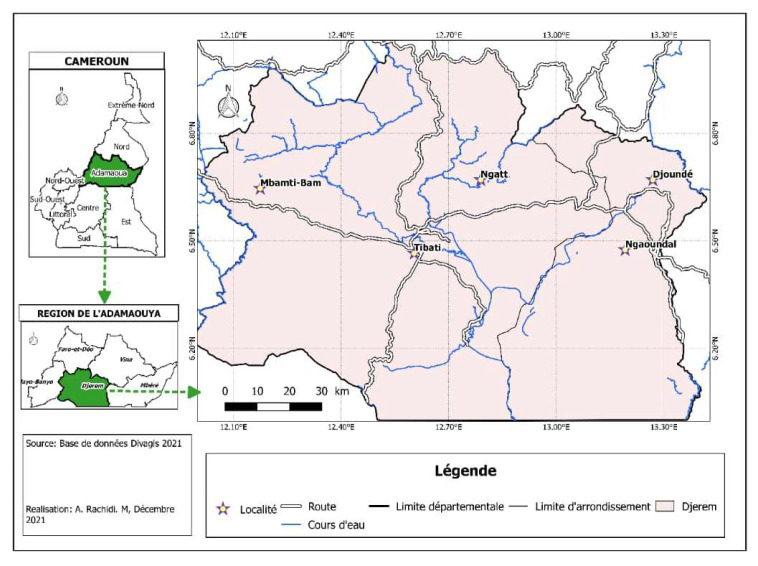
Study site (map of Djerem division).

**Figure 2 microorganisms-11-00712-f002:**
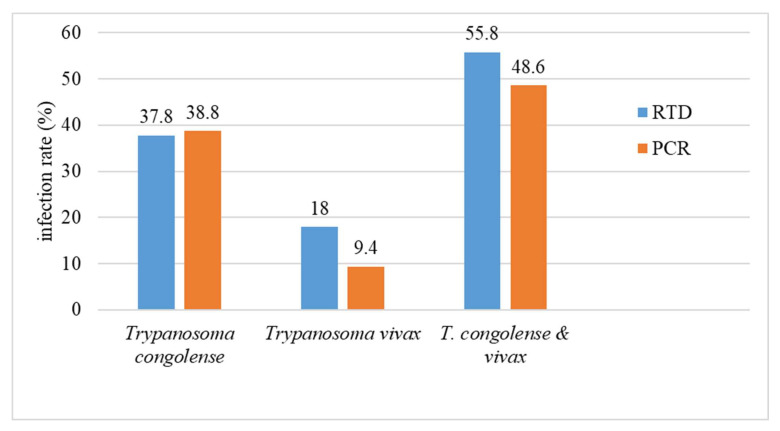
Comparison of the global *T. congolense* s.l. and *T. vivax* infection rates of cattle, regardless of the sampling villages, determined using either the PCR technique or the Very Diag Kit (RDT).

**Figure 3 microorganisms-11-00712-f003:**
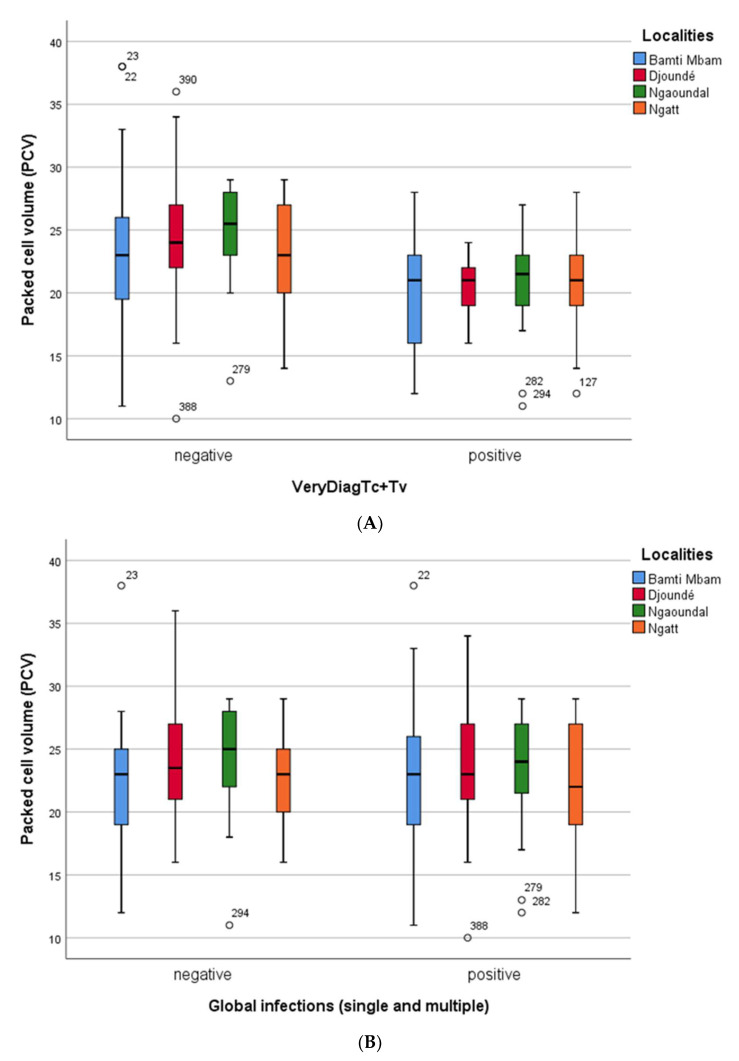
(**A**) Comparison of hematocrit rate (PCV) of trypanosome-infected (whatever the infecting species) and uninfected cattle, in the four sampling sites. Here, cattle infection was identified using the Very Diag Kit. (**B**) Comparison of hematocrit rate (PCV) of trypanosome-infected (whatever the infecting species) and uninfected cattle, in the four sampling sites. Here, cattle infection was identified using PCR technique.

**Table 1 microorganisms-11-00712-t001:** Trypanosome infection rates of cattle using Very Diag Kit.

Villages	Nb of Samples	Tc s.l. (%)	Tvx (%)	Single Infections (%)	Mixed Infections Tc/Tvx (%)	Global Infection (%)
Bamti Mbam	100	49 (49%)	17 (17%)	66 (66%)	13 (13%)	79 (79%)
Djoundé	105	35 (33.3%)	20 (19.0%)	55 (52.4%)	22 (20.9%)	77 (73.3%)
Ngaoundal	100	36 (36%)	17 (17%)	53 (53%)	22 (22%)	75 (75%)
Ngatt	100	33 (33%)	19 (19%)	52 (52%)	27 (27%)	79 (79%)
Total (Nb and mean %)	405	153 (37.8%)	73 (18.0%)	226 (55.8%)	84 (20.7%)	310 (76.5%)
χ^2^	/	7.346	0.281	5.619	6.128	1.408
P	/	0.062	0.964	0.132	0.106	0.704

Nb: Number; %: Prevalence of trypanosome infections; Tc: *Trypanosoma congolense* s.l.; Tvx: *Trypanosoma vivax*.

**Table 2 microorganisms-11-00712-t002:** Single, mixed, and global infection rates in cattle recorded using molecular (PCR) approach.

Villages	Nb Samples	Tcs (%)	Tcf (%)	Tvx (%)	Tbr (%)	Simple Infection (%)	Tcs/Tcf (%)	Tcs/Tbr (%)	Tcs/Tvx (%)	Tcf/Tbr (%)	Tcf/Tvx (%)	Tvx/Tbr (%)	Tcs/Tcf/Tbr (%)	Mixed Infection (%)	Global Infection (%)
Bamti Mbam	100	26 (26%)	4 (4%)	12 (12%)	12 (12%)	54 (54%)	3 (3%)	6 (6%)	3 (3%)	1 (1%)	1 (1%)	0	0	14 (14%)	69 (69%)
Djoundé	105	35 (33.3%)	11 (10.5%)	6 (5.7%)	3 (2.8%)	55 (52.4%)	1 (0.95%)	8 (7.6%)	2 (1.9%)	3 (2.8%)	2 (1.9%)	5 (4.8%)	3 (2.8%)	24 (22.9%)	79 (75.2%)
Ngaoundal	100	38 (38%)	8 (8%)	6 (6%)	6 (6%)	58 (58%)	2 (2%)	0	4 (4%)	0	1 (1%)	5 (5%)	3 (3%)	15 (15%)	72 (72%)
Ngatt	100	30 (30%)	5 (5%)	14 (14%)	4 (4%)	53 (53%)	0	3 (3%)	2 (2%)	0	0	0	0	5 (5%)	58 (58%)
Total	405	129 (31.8%)	28 (6.9%)	38 (9.4%)	25 (6.2%)	220 (54.3%)	6 (1.5%)	17 (4.2%)	11 (2.7%)	4 (0.98%)	4 (0.98%)	10 (2.5%)	6 (1.5%)	58 (14.3%)	278 (68.6%)
χ^2^	/	3.58	4.14	6.32	8.67	0.78	3.46	8.60	1.11	5.74	1.90	10.01	5.94	13.36	7.91
P	/	0.31	0.24	0.09	0.03	0.85	0.32	0.03	0.77	0.12	0.59	0.02	0.11	0.004	0.05

Tcs: *Trypanosoma congolense* savannah type; Tcf: *T. congolense* forest type; Tvx: *T. vivax*; Tbr: *T. brucei* s.l. (sensu lato).

**Table 3 microorganisms-11-00712-t003:** Cattle anemia level according to sampling villages.

Villages	Nb of Sample	Mean	Standard Deviation
Bamti Mbam	100	22.36	5.206
Djoundé	105	23.58	4.104
Ngaoundal	100	24.12	3.836
Ngatt	100	22.54	4.014
Total	405	23.15	4.365
F	/	3.8	/
p	/	0.01	/

## Data Availability

All data has been used in this article.

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
