# Peer review of "Trypanosome Infections and Anemia in Cattle Returning from Transhumance in Tsetse-Infested Areas of Cameroon"

_microorganisms, 2023, doi:10.3390/microorganisms11030712_

Round 1
Reviewer 1 Report
The manuscript describes a study to assess the anaemic status and the use of an immunological test and PCR-based methods to determine the infection rates of trypanosome species in cattle returning from transhumance in tsetse infested regions of Cameroon. The study is well conducted and described, however there is some lack of information that should be described and discussed. In the manuscript, the PCR methods appeared less sensitive than the Very Diag Kit, “Surprisingly, using the VDK approach, although considered as less sensitive than PCR” but there are some aspects that should be considered in the discussion. In particular, the presence of trypanosomiasis and the immunological status of the animals before the transhumance was not considered in the study. The trypanosome species can give also chronic and persistent parasitaemia, animals tested were 3 to 12 years old, it is presumable that it was not the first time that they returning from a transhumance and they could be already infested before the study. In addition, no information about previous positivity or treatments for trypanosomiasis were provided, this aspect should be also considered and discussed because the persistence of antibody positivity to an immunological test after treatment could have also influenced the difference observed in the prevalence between the immunological test and the PCR. In addition, PCR detection is a gold standard for trypanosomiasis but it can give false-negative results that may occur when the parasitaemia is very low and it is frequent in chronic infections and healthy carriers, this aspect should be discussed too.
Herein, I also reported minor revisions that should be considered:
1 the species and genus of the parasite is not always in italics in the manuscript.
2 Abstract: line 18, the objective of this work was not to check the health status of the cattle upon their return from transhumance, because no exams for other parasites or diseases were conducted. It is better to use the same sentence used in lines 83-85 “to assess the anaemic status and the use of an immunological test and PCR-based methods to determine the infection rates of trypanosome species”.
3 Line 91, march should be March.
4 Line 170 there is a “s” before villages
5 Line 175-176 there are some strange citations at the beginning and at the end of the sentence
7 Table 4: “totaux” should be “total”
8 Lines 323 and 324 there is a )
9 Line 417 “As” should be “as”
Author Response
"Please see the attachement"

Reviewer 2 Report
This study described “Trypanosome intections and anemia in cattle returning from transhumance in tsetse infested areas of Cameroon”,which indicated that cattle returning from transhumance harbored high infection rates of trypanosome infections by the detection of immunological test and PCR. Although of interest, I feel that the manuscript requires major revision.
Major comments:
1.The photos of the Trypanosome should be provided.
2. Did the author detect Trypanosome in blood by microscope?The comparison with detection of Trypanosome in blood by microscope is very important. The author should supply it.
3. The photos of Trypanosome detection by PCR should be provided.
4. What is the prevalance of tsetse flies in the regions?
5. Ethics statement should be provided.
Author Response
"Please see the attachement"

Reviewer 3 Report
The current manuscript concerns the investigation of the occurrence of trypanosome infection and anemia in cattle returning from transhumance in tsetse infested areas of Cameroon. The study is well-designed and provides valuable data however the manuscript needs some corrections.
I have the following comments which require corrections/clarifications:
1. Please highlight in the manuscript (in the introduction and/or discussion sections) what the novelty of your study compared to others.
2. Please move the information regarding the number of collected samples from the results section to the materials and methods section.
3. Line 139-146: This PCR protocol is published, there is no need to describe it.
4. Figures 3a-7: Since there is a lack of statistically significant differences there is no need to include so many plots. Please reduce the number of figures. I propose to move those removed to supplementary material.
5. Please clarify the purpose of making statistical comparisons of results between villages.
6. Please italicize pathogen names where appropriate throughout the manuscript.
Author Response
"Please see the attachement"

Round 2
Reviewer 1 Report
The authors improved the manuscript following the suggestions of the reviewers and each suggestion was considered and discussed. In my opinion, the manuscript is now suitable for publication.
Author Response
The authors improved the manuscript following the suggestions of the reviewers and each suggestion was considered and discussed. In my opinion, the manuscript is now suitable for publication.
RESPONSE: Thank you so much for all your comments, suggestions and advices. We really appreciated.
Reviewer 2 Report
I am satisfied with the author's revision.
Author Response
I am satisfied with the author's revision.
RESPONSE: Thank you very much for all your comments, corestions and advice which have improved the quelity of manuscrit.
Reviewer 3 Report
The authors addressed my comments in detail. They clarified my concerns and made suggested corrections. The manuscript has been improved and I am satisfied with these changes.
I have the following two minor points that should be corrected:
1) Please verify and correct the numbering of the suplemantary figures in the manuscript and in the supplementary materials
2) Please verify and correct Figure S1 in the supplementary materials. It seems to me that you uploaded an unfinished draft version of this figure.
Author Response
Thank you very much for all your comments, suggestions, corrections and advice which have improved yhe quelity of the manuscrit.
